# Christian Education in Colonial and Post-Independent Zimbabwe: A Paradigm Shift

**Francis Machingura** [1,*] **and Cecil Samuel Kalizi** [2]

1   Centre for Postgraduate Studies, University of Zimbabwe, Harare, Zimbabwe
2   Scripture Union Zimbabwe, Harare, Zimbabwe; samkalizi@gmail.com
*   Correspondence: fmachingura@yahoo.com

**Abstract:** Since the arrival of Christianity in Africa during the pre-colonial era, one of the main characteristics of its spread has been Christian Education (CE). The achievements made thus far by missionaries and African Christian communities were based on the Church-based Christian Education programs that were put into place by churches created by missionaries. Education, let alone Christian Education, has a key role to play in the transformation of every society. The problem is that the type of Christianity and Christian Education introduced sought to uproot Africans from their identity, culture, and language. Christian Education has a crucial role in changing the perspective of citizens to one that is Euro-centric and in promoting effective discipleship and strong doctrinal allegiance among members of mainline churches. Even though Christianity has undergone meaningful change over time, its many manifestations still survive in diverse 21st-century societies. Christianity, just like African Traditional Religion, has permeated every sphere and life of the Zimbabweans. The prospects of Christian Education to foster a positive society's transformation in Zimbabwe are great and accepted. Two types of Christianity were introduced to Africa: Afro-centric Christianity and Euro-centric Christianity.

**Keywords:** Christian Education; education for transformation; Pentecostal/Charismatic Christianity; mainline Christianity

## 1. Introduction

The Church-based forms of Christian Education, as implemented by missionary-founded churches, were critically foundational to the milestones attained so far by missionaries and African Christian communities. The role of Christian Education in fostering effective discipleship and deep loyalty to one's Church's doctrinal orientation among members of the mainline churches and in transforming citizens' worldviews to a Euro-centric Christian life seems obvious. Christianity has broadly evolved over the years, even though its various forms continue to mutate and coexist in the diverse cultures of the 21st century. Euro-centric or mainline Christianity (a form through which Christianity was introduced to Africa), Afro-centric Christianity (a form that is represented by African Independent churches), and Pentecostal/Charismatic Christianity (a form founded on teachings about the Holy Spirit) are some of the various forms of Christianity coexisting in Africa. Colonial history later included the decolonization of Africa, and some of the factors had an obvious impact or influence not only on the form of Christianity but also on Christian Education. Christian Education has a big role to play in the transformation of every society. Religions play a critical role in the character formation of society and individuals. Religions influence the identity, culture, education, beliefs, and practices of people. In the case of Zimbabwe, the coming of colonialism, Christianity, and capitalism to a people who had their own religious beliefs, practices, rituals, and education had a serious impact that has continued until today. Christianity, just like African Traditional Religion, has permeated every sphere and life of Zimbabweans. Christian churches in Zimbabwe own several schools that participate

in gospel propagation through Christian Education. The prospects of Christian Education to foster a positive society's transformations in Zimbabwe are great and accepted. A deep search and assessment of the meaning, importance, and practice of Christian Education in Zimbabwe's colonial and post-colonial eras is necessary. Christian Education has continued to play a key role in the moulding of pupils' character and contribution to peace, unity, development, and tranquility in Zimbabwe. This paper looks at the role of Christian Education in Zimbabwe during the colonial and post-colonial eras.

The impact of Christian Education on the political, social, economic, and technological spheres of African society is not only huge but also apparent. Most of the African nationalists who later became national leaders or presidents, such as Kwame Nkrumah, Julius Nyerere, Jomo Kenyatta, Nelson Mandela, Mnamdi Azikiwe, Obafemi Awolowo, Robert Mugabe, Canaan Banana, Abel Muzorewa, and Ndabaningi Sithole, were either products of Christian Education or received Christian Educational influence at some stage of their lives. Similarly, most African societies were strongly founded on Christian principles during both the colonial and post-colonial eras, a phenomenon attributable to the influence of Christian Education. It is fair to regard Christian Education as one of the most influential factors in societal transformation in Africa, particularly during the colonial era. The provision of education to indigenous Africans, mainly through missionary-established schools, and the propagation of Christianity through the same contributed immensely to the African societal transformation. The decolonization of African politics ushered Africa into a whole new trajectory of societal transformation, in which the indigenous Africans took over the governance of their nations and indeed the leadership of all spheres of society, including religion and education. Interestingly, Christianity, let alone Christian Education, thrives in schools and public spheres.

This paper looks at the historical development of Christian Education before and after the independence of Zimbabwe. The paper looks at Indigenous Knowledge Systems (IKS) in the African context and in the face of colonialism, Christianity, and Western education or culture. It shows how Christianity aided colonialism and the embracing of Western culture. However, the discourse shows the positive impact of Christian Education on African development, starting with the empowerment of African leaders who participated in and led the liberation struggle against colonialism. The paper shows how Christian Education was taught before the independence of Zimbabwe and then the paradigm shift from Christian Education to Religious Education and Family, Religious Education and Moral Education (FAREME) in postcolonial Zimbabwe in terms of Teaching and Learning Methods. The paper closes by focusing on the significance of Christian Education in Zimbabwe's Postcolonial Era.

## 2. Background of Christian Education in Zimbabwe

Defining the term Christian Education is a worthwhile position to begin with if we are going to give this paper a proper foundation. The numerous definitions of Christian Education at our disposal warrant that we propose a working definition for this paper. Frame in Philips (2023) defines Christian Education and states what it should achieve; thus, "Christian Education should seek to create a comprehensive worldview derived from biblical teachings' rather than simply teaching Bible stories or theology classes." The paper uses the above definition in its discussion as a working definition. Biblical texts, narratives or stories are societal goggles. They give testimony to the different communities that existed in ancient times, that is, the way they lived and looked at life. The narratives mostly expose the socialization processes of those communities (physically, politically, economically, religiously, culturally, and morally). The definition takes a comprehensive approach to Christian Education, in which the target is to develop in the learner a biblical worldview that leads to the cognitive development of learners. Unfortunately, this was not the same with Christian Education in Zimbabwe, as shall be shown as the discourse proceeds, where Christian Education ended up being a tool for converting indigenous Africans to Western culture. Christian Education enables people to make their faith the

foundation for all aspects of life. It equips them with critical thinking skills that are founded on Jewish Christian moral principles to enable them to make personal and professional decisions that positively transform society.

Sayes at the University of Pretoria (2021) argues that Christian Education has its roots in Jewish education, especially the Old Testament, because the entire Christianity is built on a Hebrew heritage. Any study of Christian Education must begin with the Old Testament. Dube et al. (2015, p. 76), rightfully observes that the coming of Christian missionaries ushered into the then Rhodesia and now Zimbabwe a different developmental and historical era that significantly impacted the nature of education. This did not only mark the beginning of formal education but also Christian Education in Zimbabwe. The formal education introduced was dual, one for the European settlers and the other for the indigenous Africans. Dube et al. (2015, p. 76) observe that clearly, the bedrock of the formal education introduced was Christian Education, whose objective included evangelization, colonization, and the eradication of African beliefs within a context in which many missionary organizations operated. Some of the missionary organizations were the London Mission Society, Mennonites, Roman Catholics, Lutherans, the Dutch Reformed Church, Methodists, Anglicans, the Church of Christ, Baptists, and so on. In a way, based on the above observation, missionaries designed Christian Education to displace Indigenous Knowledge Systems (IKS). Dube et al. (2015, p. 77) defines Indigenous Knowledge Systems (IKS) as "the local knowledge that is unique to a given culture or society, thus the base for local-level decision-making in agriculture, health care, food preparation, education, natural resource management and a host of other activities." Indigenous Knowledge Systems play a greater role in the socialization of people. It gives the identity, language, beliefs, and practices of others. IKS are home-grown and therefore naturally relevant to the local African context.

The coming of colonialism, Christianity, and commerce at the same time had a significant impact on the reception of Christian Education in Zimbabwean schools, that is, before and after independence. The blunder in the teaching and learning of Christian Education was to take Africans/indigenes as non-religious, traditional, backward, and primitive. Everything that identified with Africans was condemned as anti-Christian and evil, even in cases of good and progressive teachings, beliefs, practices, and customs. The resistance to missionary Christian Education in schools and missionary Christianity birthed African Indigenous Churches and African Pentecostal Christianity. Effectively, Christian Education, as introduced by the missionaries, took over as the bedrock of life and all related systems in Zimbabwe. The developments and trends of Christian Education in Zimbabwe's colonial era are important in understanding how Christian Education was begrudgingly embraced.

## 3. Developments and Trends of Christian Education in Zimbabwe's Colonial Era

### 3.1. Indigenous Knowledge System in the African Context in Contrast to Western Education

Zimbabwe's pre-colonial era was founded on IKS, or an African traditional education system, which was then replaced by the colonial formal education system. A glimpse into the IKS shows that it was well structured but misunderstood by the European settlers. Such misunderstanding is expressed by Adolphe Lous Cureau, the governor of Congo from 1900 to 1918, in Matsika (2000, p. 19), who wrote that the Negro's intellectual sphere is very limited and is almost entirely confined to the material world. He further alleged that most of the words in black or African languages described concrete objects, actions, movements, and sensations. He therefore concluded that Negros could not abstract. Cureau's way of thinking was opposed by Matsika (2000, p. 20), who perceived an African world as ruled by logic and rational thought. The African epistemological view adds the third element, that is, the spiritual or religious perspective, which takes a significant role when connecting to logic and rational thought. The African religious view influences the political, social, economic, and, of late, technological aspects. While Western epistemology only acknowledged two sources of knowledge, that is, reason (mind) and sense (experience), Oladipo (1991) in Matsika (2000, p. 20) identify religious practices such as divination,

mediumship, and witchcraft-psychic phenomena as important sources of knowledge within African epistemology. Knowledge outside the mystical perspective does not define or characterize African epistemology. This is consistent with the African worldview, in which human beings are not subject to the limitations of space but can perceive and communicate with supernatural entities.

The African philosophical worldview conditioned how indigenes interacted or engaged in whatever new or foreign cultures came to their space. The African worldview is religiously charged and spiritually connected. Matsika (2000, p. 21), proposes conversation as a way of generating knowledge in a way that is consistent with African epistemology's recognition of opinion sharing as a way of bringing about the truth. Traditionally, African men and boys would sit around a fire, particularly in the evenings after a day's hard work, to discuss all sorts of issues by way of educating one another. Women and girls would do the same while cooking or doing other chores in the kitchen. During such conversations, opinions would be raised, debated, and adjudicated upon to make sure the younger people were well equipped for life. In light of arguments by other scholars Matsika (2000, p. 141) concludes that "African philosophy has always been there in forms other than writing, that is, proverbs, stories, riddles, beliefs, myths, folktales and folk songs, customs and traditions of the people, art symbols, and socio-political institutions. African philosophy is collective; that is, it is a product of most, if not all, members of the community and is accepted by the whole community, as opposed to Western philosophy, which is a product of individual thinkers. African philosophy was the bedrock of the curriculum of IKS, that is, the education system in the pre-colonial African era."

According to Matsika (2000, p. 191), the pre-colonial African education system, which was purely IKS, sought to produce an individual with certain specific skills or traits such as military, hunting, nobility, and good character. Pre-colonial African education, that is, IKS, prepared Africans for life. The education touched a lot on life skills that enabled them to survive harsh and soft conditions. From a tender age, both boys and girls were prepared for life's challenges and opportunities. Matsika (2000, p. 192) gives characteristics of African traditional education, which are: collective and formal nature; inseparability of this education from social life both in the material and spiritual senses; and gradual and progressive nature of the African traditional education in tandem with the physical, emotional, and mental developmental processes of the child. This means that the education system was sensitive to the needs of society, and it was implemented not as preparation for life but as life itself. For the above reasons, Fafunwa and Aisiku (1982, p. 11) define education from an African traditional system perspective as

> The aggregate of all the processes by which a child or young adult develops the abilities, attitudes, and other forms of behavior that are of positive value to the society in which he lives. Education is a process of transmitting culture in terms of continuity and growth and disseminating knowledge, either to ensure social control and the rational direction of society or both.

While Faifunwa and Aisuku's definition is universal, it brings out the African traditional concept of education in a very vivid way. Education was for the good of society, just as it was also for the good of the child. It enabled the child to develop in a manner that would enable them to fit in with society.

Matsika (2000, p. 195) makes an incredibly significant observation about African traditional education, or IKS in the African context; thus, in African thought, knowledge "is not compartmentalized into theoretical and practical, intellectual and emotional, secular and sacred or materialistic and spiritual," but is holistic. He further observes that the above is consistent with the monist way of African thinking as opposed to the dualist way of Western thinking. This is one of the sources of the misunderstanding of African traditional thought and education, which led many Western scholars to take the African way of life, beliefs, practices, and philosophy as irrational. The general perspective on Africans was dismissive and judgmental.

### 3.2. Indigenous Knowledge System in the African Context, Christian Education and Colonialism

Obgo and Ndubisi (n.d., pp. 1–2) underscore the contribution of African Indigenous education, or IKS in the African context, which includes providing contextually relevant education, involving the whole community, and creating an educational environment that is consistent with upbringing. Indigenous education is perceived by Obgo and Ndubisi (n.d., p. 3) as "an insightful lifelong process of learning whereby a person progresses through predetermined stages of life from cradle to grave." The Western concept of education that came into Africa through colonialism did not recognize the existence of indigenous African education and the important role it played in bringing life and meaning to the community. The assumption was that there was no education in Africa before colonialism. The above paradigm shift meant that the Western concept of education ignored the Zimbabwean context in terms of content, particularly the expression of culture through language, social context, or the complexity of relations in indigenous communities, and cognitive culture or differences in worldviews. Mitchell (2023) carefully and accurately sums up the relationship between Christianity and colonialism in the following thoughts:

> We should admit that Christian missions played a role in the colonial process of cultural and political hegemony in Africa, India, and the Caribbean. Very often, Christian missionaries cooperated with European political rule, depending on the governmental power to maintain their status. Likewise, quite often Christian missionary schools assumed the superiority of European education, language, and ideas, and as a result, they downgraded local Indigenous culture.

Colonialism did not only involve political oppression but also religious and cultural oppression as well. Nkomazana and Setume (2016, p. 33) correctly perceive the goal of the London Missionary Society and indeed all the other missionary organizations as 'the conversion of the heathen and the promotion of their civilization.'

According to Nkomazana and Setume (2016, p. 39), civilization in this case was synonymous with European values, standards, and ways of life. In fact, "Western civilization, Christianity, commerce, and colonization were believed to be inseparable" (Nkomazana and Setume 2016, p. 38). Schmidt (2015) alleges that "Christianity played an important role in the colonization, not necessarily of the African continent, but rather of the hearts and minds of Africans, whose potential for revolt was seen as an obstacle to the imperial takeover of land and resources." While broadly Christian missionary schools cooperated with European political rule, the colonialists used them to promote European culture through an overemphasis on the use of, for example, in the Zimbabwean experience, the English language and ideas at the expense of the indigenous languages. This development resulted in the projection of white skin and European culture as superior to black skin and local indigenous culture in Zimbabwe. Intelligence became associated with one's capacity to communicate through the English language and articulate Western ideas, irrespective of their relevance or lack thereof to the local context.

The possibility of cross-pollination of cultures and ideas was very remote and limited, as the local indigenous culture and ideas were automatically labelled inferior through Christian Education. What made it worse was the fact that everything European, including names, was accepted as Christian while everything African was denigrated as pagan. Traces of such a development were still evident in Zimbabwe 43 years after independence. For example, it was common for indigenous people to address someone they perceived as affluent as *murungu* in Shona or *khiwa* in Ndebele; both indigenous words in Zimbabwe mean white man. Furthermore, it was a common trend to find indigenous people who were neither good in their local language nor English language because of the education system's emphasis on a foreign language. However, Mitchell (2023) also acknowledged that sometimes Christian missionaries opposed oppressive colonial rule by promoting the rights of the colonized and supporting the liberation struggle. To that effect, the Christian missionaries later "sought to work within the language, customs, and cultural structure of indigenous peoples and, equally, sought to downplay their Westernized understanding

of matters" (Mitchell 2023). Notwithstanding the above, Christian Education contributed significantly to the colonial agenda of denigrating the local culture and ideas.

### 3.3. The Positive Impact of Christian Education on African Development

Schmidt (2015) observed that much as Christianity played a significant role in destroying African culture and identity, it later played an equally crucial role in its restoration, as authenticated by the fact that most African nationalists were products of missionary education in terms of giving them the needed superior logic of politically helping the liberation of their people. The same Christian faith that had been abused by some missionaries and colonialists to pacify Africans also enlightened them to realize that all human beings were the same before God. The Christian faith became an amazingly effective foundation for mounting formidable resistance against colonial rule in Zimbabwe and, indeed, in Africa. Schmidt (2015) correctly observes that Christianity was not the only factor that led to both the colonization and the decolonization of Africa, even though it played a significant role. Many forces, including religion beyond just Christianity, were at play in both processes.

### 3.4. A Paradigm Shift from Christian Education to Religious Education and FAREME

Masengwe and Dube (2023b) narrate the history of Christian Education in the Church of Christ in Zimbabwe (COCZ) and observe that COCZ used Christian Education to preserve its doctrines until the introduction of Religious Education (RE) and later Family, Religious and Moral Education (FAREME) in schools. Religious Education and FAREME were multi-faith syllabi through which different religious traditions like Judaism, Islam, African Traditional Religion (ATR), Hinduism, and Buddhism were taught without proselytization. Before the introduction of Religious Education and FAREME in mission schools, Christian Education was exclusively Christian in terms of content, to the effect that it was referred to as Bible Knowledge or scripture. The above development posed a great challenge to missionary-founded churches' endeavour to promote Christian Education through their mission schools. This was particularly true considering Pethtel (2011, p. 2)'s view of Christian Education, that is, "the belief that all learning and truth is ultimately from God and must be understood through a close understanding of the Bible." The definition equates Christian Education with Bible Knowledge, whose ultimate end is both evangelism and discipleship. However, contrary to the missionary-founded churches, such an end was described by Zimbabwe's education authorities as proselytization and exclusive to the other religious faiths in Zimbabwe.

The authorities would rather have learners exposed to various faiths without coercion and allow them the opportunity to choose a faith whenever it is convenient for them. To illustrate the above, MoPSE (2015a, p. 36) highlights one of the principles governing FAREME as "the need for learners to appreciate the diversity and practices of various religions practiced in Zimbabwe." The premise for the principle was the thought that "religion had a pervasive influence in our society, including the lives of learners" (MoPSE 2015a, p. 36). The MoPSE's position was founded on the fundamental human rights and freedoms enshrined in Zimbabwe's constitution, as follows:

> Every person has the right to freedom of conscience, which includes—freedom of thought, opinion, religion, or belief; and freedom to practice, propagate, and give expression to their thought, opinion, religion, or belief, whether in public or in private, and whether alone or together with others. (Zimbabwe 2013, p. 30)

While religions were still allowed under Zimbabwe's laws to establish institutions like schools where religious instruction could be given, the examinable curriculum was designed, provided, and monitored by the MoPSE. Under such a curriculum, Bible Knowledge could only be accommodated under co-curricular elements that were non-examinable and not mandatory. The Bible Knowledge curriculum was so limited and frustrating.

*3.5. The Christian Education Teaching Methods during the Colonial Era*

Ndlovu in Muhamba (2019, p. 25) argues that "there was no clear government policy on how Religious Education was to be taught in both missionary and government schools" during the colonial era in Zimbabwe. The above was helpful in the exploration of the methods of teaching and learning Christian Education in the colonial era in Zimbabwe. Each institution had the privilege of designing its curriculum to suit its background and the beliefs of the missionary Church. The first national education curriculum came into being as an initiative of John Cowie, the then secretary for education in Rhodesia; the mission schools continued to use their self-designed curricula (Muhamba 2019, pp. 25–26). The specific school's Christian denomination and doctrine determined the content of Christian Education and methods of teaching and learning. However, in government schools, Christian Education was called Bible Knowledge, and the study of other religions in both mission and government schools was excluded. Wiebe (2015, p. 16) identifies some of the methods that were used for the teaching and learning of Christian Education during the colonial era in the Belgian Congo, and they were also used in Zimbabwe.

The teaching methods were the simple memory discipline, the textbook method, and the lecture method. The memory discipline included the memorization of scriptural verses, creeds, statements of faith, and moral codes. While this method is still in use today, during the colonial era it lacked the creativity that should make it an exciting rather than a boring exercise. The textbook method was one in which the teacher and their learners went through the content in the textbook together while the teacher explained it. The lecture method involved the teacher verbally delivering content to the learners while the learners listened attentively. Both the textbook and the lecture methods had either the barest minimum or no learner participation in the process (Wiebe 2015, p. 16). The methodology was founded on the philosophy of hegemony and dominance over the African indigenous people by the colonial powers. The teaching and learning methodologies promoted rote learning and the attitude of looking for employment after schooling. This type of Christian and Religious Education celebrates Western culture at the expense of other cultures.

Missionaries believed that Religious Education should be dogmatic and confessional (UNISA n.d., p. 49). For the Christian missionaries, Christian Education was designed to promote evangelism and to transform Africans from traditional life and religion to Christianity. The teaching and learning methods were therefore designed to fulfil such a mandate. The methods were purely instructional rather than facilitative. When the colonial government was set up in 1890, the missionaries made it clear that the education of Africans would continue to comprise literacy and religious instruction (UNISA n.d., p. 49). The missionaries' position aligned very well with that of the colonialists, who sought to produce cheap labour through their education system. Then, Christian Education was presented as Scripture or Religious and Moral instruction (UNISA n.d., p. 49). The Graham Commission on Native Affairs in UNISA (n.d., p. 53) recommended that African education during the colonial era should concentrate on three aspects: literacy, religion, and practical training. The missionaries' focus on their education of Africans was evangelism, while the colonial Government's focus was the production of cheap labour. Both Christian missionaries and the colonial government did not educate Africans neutrally, but they equipped them to promote their business enterprises. Christian Education was the hidden curriculum of colonialism. What manifested in the school as colonial education manifested in the Church, with the emphasis on going to heaven rather than striving to gain wealth in the here and now. All forms of ill-treatment by the colonial settlers were overlooked, with an emphasis on the reversal of pain in the hereafter. Christianity as a religion became a 'lullaby religion' for those in pain in the afterlife.

The enterprise of the Christian missionaries was the propagation of the Christian faith. All the knowledge Africans needed to acquire to pursue the enterprise of the missionaries was literacy, particularly the ability to read the Bible and Christian doctrine, which varied from one Church to the next. The enterprise of the colonialists was capitalism, as expressed through Agriculture, mining, manufacturing, and the rest of commerce. Africans having

been earmarked for manual and other semiskilled labour activities, the highest education level they needed, according to the colonialists, was practical training in areas such as building, carpentry, metal work, basic farming, and so on. This sufficiently prepared and equipped Africans as employees. Scripture, or Religious and Moral Education, played a very key role in the colonial business matrix. Colonial Education disempowered Africans and their traditions. The entire system and its education prepared Africans to work, become employees, and never become employers. The education helped produce an honest, trustworthy, dependable, respectful, and submissive labourer who promoted and protected the master-servant relationship between the colonialist and the colonized. Christian Education, Religious Education, and Moral Education during the colonial era were designed to promote the interests of the colonialists in Zimbabwe.

The motives for Christian missionaries and Rhodesia's colonial government educating Africans were clear. Zvobgo (1981, p. 13) identifies two historical salient features of the Rhodesian education system, namely, racial segregation and the exploitation of most Africans. The two features are overly critical in the discussion when the role of Religious Education in national development during the colonial era is assessed. The features suggest that the African majority was marginalized from Rhodesia's developmental plans, even though they contributed much-needed labour to the nation. Kallaway (2020, p. 33) asks a very pertinent question:

> was colonial education about creating African Christians/African workers/African subjects/citizens who were to be the vanguard of social, economic, and political modernization and perhaps Westernization, or was the role of schools and missions to prevent such modernization and radicalization by facilitating a more productive life in the land for peasant farmers and contented "tribesmen" or educated indigenes who would not threaten the colonial order?

Shizha and Kariwo (2011, p. 17) respond to Kallaway's (2020, p. 33) questions by asserting that "both missionaries and educational administrators" in Rhodesia "introduced an educational system for Africans that was designed to overtly and explicitly marginalize Africans and strengthen and sustain African domination." They achieved this by underfunding African education and adequately funding (in abundance) European education in the same nation. Wa Thiongo, an African writer, as cited by Shizha and Kariwo (2011, p. 13), laments the following, considering the contribution of Religious Education to development during Africa's colonial era:

> Religion is not the same thing as God. When the British Imperialists came here in 1895, all the missionaries of all the churches held the Bible in their left hand and the gun in their right hand. The white man wanted us to be drunk with religion while he, in the meantime, was mapping and grabbing our land and starting factories and businesses on our sweat.

Wa Thiongo's observation was an underlying principle in the operations of the missionaries, whether each missionary knew it or not. This was because they were funded by their sending missionary organizations, which were headquartered in either the colonizing countries or their allies. The letter by Leopold II (Letter from King Leopold II of Belgium to Colonial Missionaries 1883) to missionaries in Congo says it all (https://www.fafich.ufmg.br, accessed on 10 January 2024), Reverends, Fathers, and Dear Compatriots:

> The task that is given to fulfil is very delicate and requires much tact. You will certainly go to evangelize, but your evangelization must inspire above all Belgium interests. Your principal objective in our mission in the Congo is never to teach the niggers to know God, this they already know. They speak and submit to a Mungu, one Nzambi, one Nzakomba, and what else I don't know. They know that to kill, to sleep with someone else's wife, to lie, and to insult is bad. Have the courage to admit it; you are not going to teach them what they know already. Your essential role is to facilitate the tasks of administrators and industrials, which means you will interpret the gospel in the way that will be best to protect your interests in

that part of the world. For these things, you must keep watch on disinteresting savages from the richness that is plenty in their underground. To avoid that, they get interested in it, and make you murderous competition and dream one day to overthrow you. Your knowledge of the gospel will allow you to find texts ordering, and encouraging your followers to love poverty, like "Happier are the poor because they will inherit the heaven" and, "It's very difficult for the rich to enter the kingdom of God." You must detach from them and make them disrespect everything which gives courage to affront us. I refer to their Mystic System and their war fetish—warfare protection—which they pretend not to want to abandon, and you must do everything in your power to make it disappear. Your action will be directed to the younger ones, for they won't revolt when the recommendation of the priest is contradictory to their parent's teachings. The children must learn to obey what the missionary recommends, who is the father of their soul. You must singularly insist on their total submission and obedience, avoid developing the spirit in the schools, and teach students to read and not to reason. There, dear patriots, are some of the principles that you must apply. You will find many other books, which will be given to you at the end of this conference. Evangelize the niggers so that they stay forever in submission to the white colonialists, so they never revolt against the restraints they are undergoing. Recite every day "happy are those who are weeping because the kingdom of God is for them".

This means that their hermeneutics and theologies were influenced deliberately leading to an overemphasis on going to heaven at the expense of life in the here and now.

The Christian teachings during the colonial era prepared Africans for life in heaven while they were marginalized by colonialists in their own country. Such teachings, coupled with the biblical demand for the servant to submit to the master (Colossians 3; 22–24; Ephesians 6: 5–9; 1 Pet 2: 18–20) as applied to the relationship between the majority of Africans, who were largely employees, and the colonialists, who were largely employers, point to a hermeneutic of convenience. It is such hermeneutics and biblical analysis that did not benefit the broader Africans.

## 4. Christian Education in Postcolonial Zimbabwe: A Paradigm Shift

### 4.1. A Paradigm Shift in Christian Education from Colonial to Postcolonial Eras

There was a change in basic assumptions in the Christian Education narrative in postcolonial Zimbabwe, which was influenced by the decolonization process. This is because Christian Education during the colonial era in Zimbabwe was never neutral. Instead, Masengwe and Dube (2023a, p. 128) argue that the Christian Education curriculum, just like the entire education system of the time, was heavily influenced by the colonial powers' interests in Christianity, colonialism, and commercialization. The colonial powers' interests were founded on the faulty premise that "the most valuable knowledge and the most valuable ways of teaching and learning come from a single European tradition (Charles in Masengwe and Dube 2023a, p. 128). Muhamba (2019, p. 28) notes that after independence in 1980, the Zimbabwe government introduced a religious policy that guaranteed the freedom of any person to practice their faith publicly or privately. The policy provided a foundation for a new approach that replaced Christian Education (CE) with Religious Education (RE), later Religious and Moral Education (RME), and more recently, Family, Religious, and Moral Education (FAREME). Before the introduction of the new curriculum by MoPSE, through which FAREME came, RE and RME were taught up to the second year of secondary education while at ordinary and advanced levels, Bible Knowledge and Divinity were taught respectively. Bible Knowledge and Divinity were still exclusively Christian in content and aligned with the Christian Education approach of the colonial era. Both Religious Education and Religious and Moral Education accommodated other religions apart from Christianity while emphasizing certain core values, such as love, with the view of building the character of the learner. While FAREME still did the same,

it had an added dimension of family to help learners value the importance of various religions (s) in a family.

The new curriculum is not limited to focusing on Christianity as the only religion, but other religions are roped in to help students appreciate diversity and build tolerance in societies and communities. Modern curriculum developers have realized the importance of knowledge and information transfer about different religions that exist in the world as well as nationalities. This builds unity, peace, and tolerance, which are critical ingredients for national development. The Zimbabwean pre- and post-colonial Christian Education was good at contributing to the moral development of pupils and society. The primary school level of grades 3 to 7 FAREME as a learning area is described by the Ministry of Primary and Secondary Education (MoPSE 2015b, p. 3):

> Designed to promote in learners an awareness and appreciation of different religions practiced in Zimbabwe. The learning area seeks to develop a sense of family cohesion, unity, moral uprightness, inclusivity, and tolerance among citizens with acceptable behaviors and values (*Unhu/Ubuntu/Vumunhu*).

Three aspects are emphasized in the current FAREME subject: awareness-raising and developing in the learner a sense of appreciation for the different religions in Zimbabwe, a sense of family cohesion, and the inculcation of core values that will help to preserve the humaneness of our society. While the development sounds positive in terms of national development, it raised a challenge for the Christian community in Zimbabwe. How else could it promote its tenets through Christian Education outside the Church building or premises? The topics included in the syllabus were family (concept of family, family and religion, family and community, social and emotional learning), religion (concept of religion, Indigenous religion, Christianity, Judaism, and Islam), moral values, and finally religion and health. The syllabus was not designed to promote the Christian faith but to show how the different religions contribute to the moral development and prosperity of the nation.

Unlike in the colonial era, when Christianity was the religion of the colonizers or the ruling class and the de facto national religion without question, in the postcolonial era, the ruling class was a mixed group with allegiance to both ATR and Christianity. The key government offices and institutions in the capital city of Harare, as designed by the colonial government, still depict Christianity as the de facto national religion. The parliament, until very recently, was housed in the same building as the Anglican Church Cathedral and in the same vicinity as the headquarters of the defence forces, the MoPSE headquarters, and indeed the office of the president, which would have been the office of the Rhodesian Prime Minister. African Traditional and indigenous religions were publicly denigrated for becoming 'rural' religions. Most urban dwellers cannot publicly associate with African Indigenous religions as compared to Christianity or possibly most rural dwellers. The other missionary-founded churches, like the Methodist Church in Zimbabwe, Presbyterian, Dutch Reformed, Baptist, and Roman Catholic, also have their Church premises in the prime spots of the Harare city centre. Was this arrangement meant to create an all-time impression of Christianity as the exclusive religion for the nation of Zimbabwe, irrespective of who was in power? More importantly, Zimbabwe's postcolonial governments under both Robert Mugabe and Emmerson Mnangagwa comprised people who identified with one Church or the other despite their allegiance to ATR beliefs as expressed in their speeches and practices from time to time.

Hence, Zimbabwe's school curriculum from 2015 to 2022 was underpinned by an indigenous philosophy, *unhu/ubuntu/vumunhu*, indigenous words that mean humaneness (Bhurekeni 2020, p. 109). It is clear that, no matter how much postcolonial Zimbabwe associated Christianity with colonialism, most of its population, including those in high government positions, whether genuinely or otherwise, identified with Christianity. This was because of the operations of the different churches in Zimbabwe, which took over a century. The colonial conversation with Christianity is the rationale for the multi-faith approach to education, as expressed through Religious Education, Religious and Moral Education and FAREME in postcolonial Zimbabwe. Unlike in the colonial era, when

Christian Education had full government support, in the postcolonial era, it was challenged to thrive in an environment where the government promoted freedom of religion or belief and discouraged the proselytization of learners by any religion in schools. The Zimbabwean government's position to invite other religions (Islam, ATR, Judaism, and Hinduism) led some Christian organizations or churches to remove the teaching and learning of FAREME because of their stance against the teaching and learning of other religions. Scripture Union and some churches still use teaching and learning or school spaces to propagate the Christian gospel or organize special meetings for the same purpose, something not possible for other religions such as Islam.

### 4.2. Need for the Church's Paradigm Shift in Platforms and Systems for Christian Education

Scripture Union, a Christian organization whose mission was to evangelize and disciple children and youth through schools and related strategic alliances that operated in Zimbabwe since 1945, was not amused in 2016 when the MoPSE instructed it through a meeting with the Zimbabwe Heads of Christian Denominations (ZOHCD) to halt operations (The Sunday Mail 2016). According to an anonymous MoPSE official quoted by The Sunday Mail (2016), the Zimbabwe government wanted to adopt a multi-faith system in schools, and the activities of the Scripture Union in schools would only be permitted if they were found to be in line with the new curriculum. Scripture Union activities were some of the CE platforms in schools, which were firmly established for many years because the colonial era promoted Christianity. The homework, not just for the Scripture Union but for the whole Church in postcolonial Zimbabwe, was to innovate platforms and systems for Christian Education in schools despite the multi-faith environment the government was currently promoting. This was probably the time for Scripture Union, other Christian organizations, and the church to consider setting up community interdenominational platforms outside schools to engage children and youth in Christian Education. The chaplaincy system of giving Christian Education to school learners at various community facilities like churches and community halls as modelled by Scripture Union Australia was a concept worth studying, contextualizing, and applying in Zimbabwe (Menzies 2022, p. 2). While the Scripture Union clubs were still running in Zimbabwe schools, the school timetable was too congested in line with the new curriculum introduced in 2015, leaving little time, if any, for Scripture Union activities.

Other Christian Education initiatives that transcended school boundaries included the effective use of social media platforms, national and community radio stations, and the production of child-friendly Bible reading materials for free distribution. It was a season for the Christian community in postcolonial Zimbabwe to think creatively if Christian Education were to survive the challenges of the multi-faith approach to education. The positive factors supporting the above-suggested initiatives for promoting Christian Education were the growth of technology, the availability of smartphones and other computer gadgets, and the liberalization of the airwaves, leading to an increase in the number of national and/or community radio and television stations. According to Doyle et al. (2021), who surveyed youth aged 13–24 in 5 communities in urban and peri-urban Harare and Mashonaland East, Zimbabwe, 62.6% reported owning a mobile phone and 4.3% reported having access to a shared mobile phone. Out of the owned or shared phones, 85.3% were smartphones, and 64.5% had internet access. The survey results demonstrated an increase in young people's access to smart mobile phones and the internet. Assuming that the trend was national, the Christian community had a huge opportunity to churn out Christian Education through social media platforms like YouTube, TikTok, Facebook, Instagram, and WhatsApp.

### 4.3. Need for the Church's Paradigm Shift in Christian Education Teaching

Unlike during the colonial era, when the teaching methods for Religious Education and Christian Education were confessional and dogmatic, the postcolonial era demanded different methods (Machingura and Mugebe 2015). Postcolonial Zimbabwe constitutionally provided freedom of religion and belief (Zimbabwe 2013, sec. 44–48); and the underpinning

of the educational curriculum on the indigenous philosophy, *ubuntu*, which was rooted in ATR, were forms of confrontation to Christianity as much as they were forms of confrontation to colonialism. They also demanded a shift in methods of teaching Christian Education. According to Masengwe and Dube (2023a, p. 128), the philosophy already mentioned summarised Zimbabwe's demand for dialogue between the Euro-centric, Biblio-centric, and Christo-centric Religious Education curriculum of the colonial era and other religious beliefs obtained on Zimbabwe soil, particularly ATR. Unfortunately, the Euro-centric, Biblio-centric, and Christo-centric exclusive approach to Christian Education continued in the postcolonial era, resulting in human products that had many cultural experiences they could not explain from a biblical perspective. Hence, Masengwe and Dube (2023a, p. 128) advocate for the need to remove the oppressive systems of education and replace them with democratic and accommodating pedagogical systems.

Masengwe and Dube (2023a, p. 129) further argue that the decolonization of Christian Education and Religious Education curricula is critical because religion cuts across culture, education, technology, and the people's ways of life, which depict the identity and dignity of a person. Issues of education contribute to people's esteem, integrity, and development. Such decolonization is incomplete without a shift in methods. Masengwe and Dube (2023a, p. 131) perceive democratic and accommodating pedagogical systems as teaching methods that allow the learner the opportunity to choose, to be innovative, and to be entrepreneurial. Such methods are also applied to Religious Education and Christian Education. The role of the educator is not to impose knowledge but to facilitate learning and challenge the learner to discover knowledge and use the knowledge in ways that are productive for the good of society. How can the facilitators of Religious Education and Christian Education in postcolonial Zimbabwe help a learner to personally know God or engage in discourses about the divine in ways that are relevant to their context? How can a facilitator of Religious Education and Christian Education in a postcolonial Zimbabwe classroom setting or environment help a learner to know God personally in ways that help them maintain an African identity, integrity, and dignity? Undoubtedly, there is a great need for teaching approaches that equip learners with Bible reading and interpretation skills, as opposed to regurgitating Eurocentric biblical doctrines. The need to facilitate learning by the Religious Education and Christian Education facilitators so that the learner develops the capacity to generate ideas instead of the elitist approach where the teacher is the source of all the ideas is apparent. Such teaching and learning methods apply from the lowest to the highest levels of education, as well as Church and Scripture Union-related programs.

### 4.4. The Significance of Christian Education in Zimbabwe's Postcolonial Era

Christian Education, Religious Education, and Moral Education (ME) in postcolonial Zimbabwe are very much necessary, just as during the colonial era, with a different approach. While FAREME, a combination of family-related education, Religious Education, and Moral Education, has a role to play in postcolonial Zimbabwe, it cannot replace Christian Education. This is particularly important considering the demographics, which show that 85% of Zimbabwe's population professes to be Christians, yet violence, rape, suicide, moral decay, poverty, femicide, unemployment, human rights abuses, and neglect are highly prevalent (Inter-Censal Demography Survey Report in Masengwe and Dube 2023a, p. 127). The fact that most Zimbabweans profess to be Christians yet immorality and evil are common is a problem that the nation, particularly the Christian community, must worry about. Masaka (2011, p. 3) describes the period under the pre-government of the Government of National Unity (GNU) of 2009 to 2013 in Zimbabwe as characterized by moral paralysis, which included the implementation of wholesale price controls by the government, lawlessness, a chaotic and violence-ridden land reform program, speculative pricing of commodities, black market trading, and various forms of corruption.

While the democratization of the economy as prescribed by Masaka (2011, p. 3) could be helpful, the need for moral sanctity in Zimbabwe was apparent. There seemed to be such an affinity for immorality that corruption had become endemic. Former Zimbabwe's Reserve Bank governor, Gideon Gono, in The New Humanitarian (2006) said,

> If, as a nation, we do not resolutely stamp outgrowing corruption, especially among us people in positions of authority and influence, we will soon discover, too late, that policy formulation, implementation, monitoring, and decisions have been based on self-interest, racial overtones, and regional and tribal considerations at the expense of the national good.

With 85% of Zimbabwe's population professing to be Christians, it defies all logic that the nation is suffering from endemic corruption that has stalled national progress and development. Emmerson Mnangagwa, the president of Zimbabwe, pledged in his inaugural speech in November 2017 that,

> As we focus on recovering our economy, we must shed the misbehaviour and acts of indiscipline that have characterized the past. Acts of corruption must stop forthwith. Where these occur, swift justice must be shown for every crime, and other acts of economic sabotage can only guarantee ruin to perpetrators. We must aspire to be a clean nation, one sworn to high moral standards and deserved rewards.

The statement by President Mnangagwa was an admission of the existence of corruption in Zimbabwe and a pledge to deal with it decisively. Whatever mechanisms the state put in place to deal with immorality as exemplified by widespread corruption in Zimbabwe would not be effective without a change of behaviour by citizens. Some of the ways to foster behaviour change in postcolonial Zimbabwe include teaching and learning Christian Education, Religious Education, and Moral Education which are critical in building the moral conscience of society.

The influence of Christianity in the teaching and learning of Religious Education in postcolonial Zimbabwe would have been much stronger if the church had contributed to the formulation of the new curriculum rolled out between 2015 and 2022. It was the one through which FAREME was introduced to replace Religious Education, Christian Education, and Moral Education. Whether the Zimbabwean Church was engaged genuinely by the MoPSE during the formulation process of the new curriculum is a debate beyond the scope of this paper, but suffice to highlight that even after its introduction, the Church did not meaningfully engage the government about the development. The state of the Church in Zimbabwe has evolved significantly since Zimbabwe's independence in 1980, and it was such that Christian Education was no longer a priority as it was during the colonial era. According to the Inter-Censal Demographic Survey in the Research and Advocacy Unit (2018, p. 2), 84% of the population is Christian. Out of the 84%, 34% belonged to African Independent Churches (AIC), 20% were Pentecostals, 16% were Protestants, and of the remaining 30%, 8% were Catholics, while the rest were accounted for by other religions. Most of the AICs and Pentecostal churches did not have schools and therefore did not have the culture of engaging with civic issues. This left Protestants and Catholics in the minority, even though they had schools and a culture of interacting with civic issues. As a result, the influence of Christianity in the teaching and learning of Religious Education in postcolonial Zimbabwe had been significantly watered down. Other religions are now part of the lion's share of the new curriculum. The above-painted picture left such organizations as Scripture Union with a huge challenge of filling in the gap created by the status of the Church in Zimbabwe, particularly in terms of promoting Christian Education in the country. Christian Education can be taught and learned not for examination but just to contribute to the moral development of pupils and youth. What matters is not whether Christian Education is examined or not, but its existence and contribution to the fight against the moral evils of society.

Postcolonial Zimbabwe saw the breaking down of racial segregation in education, the building of many schools, and the provision of education for all. Unlike in the colonial era, where Christian Education and Religious Education were largely used to promote colonial interests, the postcolonial agenda was that of restoring the dignity of the indigenous African people. Kanyongo (2005, p. 66) highlights that Zimbabwe's postcolonial education system was distinguishable from the colonial one by the unification of the two separate education systems, one for whites and the other for blacks, to remove the anomalies and inequalities. Inequalities included more than 20 times the funding for the education of whites in comparison to blacks. The massive expansion of the education system not only provided the indigenous people with access to education but also created employment as the demand for trained teachers grew as well. The socialist approach is a major highlight of postcolonial Zimbabwe because it satisfies the needs of most people. Unfortunately, due to exogenous and endogenous factors, the economy did not grow enough over the period to sustain the social services that Zimbabwe was giving to most of its citizens (Kawewe and Dibbie 2000, p. 82). The teaching and learning of Religious Education in the post-colonial context had a different focus, namely, the restoration of the dignity and identity of indigenous people, national development in a holistic manner that genuinely includes the indigenous people, and participation of the indigenous people in job creation and industrialization. The focus, however, was still a work in progress, and sadly, 43 years after independence, the ideals of national development were yet to be realized.

## 5. Conclusions

Christian Education has a long history in Zimbabwe, beginning with the introduction of Christianity by the missionaries before colonization, the introduction of formal education for indigenous people by the missionaries, and the introduction of a dual education system for whites and blacks. However, before the introduction of formal education by missionaries in Zimbabwe, the indigenous people had their own knowledge and ways of passing it on to each other. In other words, they had their own education system, which the missionaries and colonial settlers disregarded. Christian Education and Religious Education during the colonial era served the interests of the colonialists, even though missionaries sometimes promoted the rights of the indigenous people. The methods of teaching and learning Christian Education and Religious Education during the colonial era were confessional and dogmatic while aiming to proselytize the indigenous people. While such methods promoted Christianity through the proselytized indigenous people who would help the missionaries to evangelize, they also promoted the interests of the colonialists, who were interested in cheap labour and yet comprised of honest, trustworthy, respectful, and reliable people. The postcolonial era was a different context, demanding different approaches to Christian Education and Religious Education. Democratic and accommodating teaching and learning methods were most appropriate, as they would allow dialogue between Christianity and other religions to enable Africans to know God in ways that were relevant to them. Having noted that religion cuts across everything, it was imperative to conclude that the teaching of Christian Education and Religious Education remained relevant in the postcolonial era as they contributed to much-needed citizen behaviour change and national development. We expect the teaching and learning of Christian Education and FAREME to move with the times in terms of utilising ICT skills so that learners enjoy their teaching and learning.

**Author Contributions:** Conceptualization F.M., Methodology F.M., Investigation C.S.K., Writing—original draft presentation C.S.K., Writing, review and editing F.M. All authors have read and agreed to the published version of the manuscript.

**Funding:** This research received no external funding.

**Institutional Review Board Statement:** Not applicable.

**Informed Consent Statement:** Not applicable.

**Data Availability Statement:** No new data were created or analyzed in this study. Data sharing is not applicable to this article.

**Conflicts of Interest:** The authors declare no conflict of interest.

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
