# Peer review of "Christian Education in Colonial and Post-Independent Zimbabwe: A Paradigm Shift"

_religions, doi:10.3390/rel15020213_

Round 1

Reviewer 1 Report

Comments and Suggestions for Authors

Repetition of some words like understandably may indicate that AI was used in writing. This calls for closer scrutiny of ideas being presented and verification with sources of data.

Comments on the Quality of English Language

English is not that bad but some sentences are unnecessarily long.

Author Response

Thank you for the reviews. We enjoyed addressing the concerns raised. We went through the entire document and addressed the gaps (language and grammar and spellings).  Two language editors from our university went through the paper, and they indicated areas that needed corrections. Long sentences have been shortened but maintaining the discourse.  Repeated words have been addressed. We are satisfied with the paper. Please take note that we have introduced sub-headings in the paper for clarity. We hope that the corrections have improved the quality of the paper.  We look forward to your feedback. 

Reviewer 2 Report

Comments and Suggestions for Authors

The author needs to work on the references, the statement of the problem, and also shorten the abstract. See attached repott

Comments on the Quality of English Language

The English is good, just a few corrections are editions in the attached documents that needs attention. 

Author Response

We have worked on the statement of the problem, references and shortened the Abstract. We are now happy with the paper as we managed to addressed your concerns. We addressed the raised concerns and made corrections. 

Reviewer 3 Report

Comments and Suggestions for Authors

As a short, comprehensive description of Christian education in Zimbabwe, the article is fine.  In addition, for a non-specialist in the history of Christian religious the article is a good, quick introduction.  if that fits the editors’s purposes for this issue, the essay will be a contribution  

I personally wish the essay were more nuanced and focused.  One of its major problems in presentation is that paragraphs are much too long.  This confuses the reader as the argument is developed. For example, the short conclusion could easily be 3 or 4 paragraphs, instead of one.  The summary would then be clearer  

Let me suggest three areas where the argument could be improved:

First, the definition of Christian education: instead to providing a focused definition, the author quotes many authors.  It feels like view is placed on top of view.  

I’d rather have the author provide a distinct definition on which the essay is based. The author primarily focuses on philosophy and  purposes of CRE. I’d like that stated. I think any good definition of CRE mentions attention to sacred texts, theology, practices, and religious rituals. CRE has many forms. Most forms have both commitments to teaching a view of Christian faith and its implications for persons and social/cultural contexts.

Secondly, the introduction to African traditional education was clear and informative.  It would have been helpful to mention how it helped persons and groups develop and pass on heritages and deal with change. I think some brief discussion about how it contrasted with Western practices for what are called both humanities and sciences would be helpful.

Third, I have no question colonial Christian education sought to teach a religious perspective, challenge traditional African cultures, and provide control for colonial powers and their industries.  This is well suggested in the article, but how that education also empowered contemporary leadership and practices is barely mentioned — that is, about how that education itself had conflictual, challenging, and transformative elements in its practice. 

Finally, in the conclusion, I’d like to see some discussion/ proposal for next steps.

If the editors want a quick, helpful and focused introduction of the history of CRE in Zimbabwe, this article works.  Nevertheless, it needs edited and paragraphs need to develop the argument. I would prefer a more focused essay — one with a clear, defined purpose that builds on and challenges the past, describes the present, and offers directions. 

Comments on the Quality of English Language

Just like the abstract goes on and on, so do paragraphs. The focus of the essay would be enhanced by significant reformatting. 

Author Response

The article is now more focused after our university language editors went through the paper. Paragraphs have been shortened. Definition addressed and working definition of Christian Education adopted. Introduction has been revisited and made clear.Essay has shown how contemporary leaders and the society at large benefit from Christian Education. Essay has been focused and reformatted. Thank you for the constructive comments. 

Round 2

Reviewer 3 Report

Comments and Suggestions for Authors

Much improved. The sub-headings make it much more readable. As an introduction to the history of CE in Zimbabwe, the article is very helpful. It also advocates for potential directions. The article will fit well in the collection of essays the editors plan.

Advice for the future: Clarify the essential thesis for an essay. Develop the essay with this in focus. Too many articles become confusing when too many ideas are included and too many tangents followed.

Also strong introductory paragraphs at the beginning that draw the reader in and focus interest are crucial!  Much improved. 

Author Response

All corrections have been highlighted in yellow. We have addressed all the corrections. Thank You so much for the recommendations. 
